# Bone Morphogenetic Protein-4 Impairs Retinal Endothelial Cell Barrier, a Potential Role in Diabetic Retinopathy

**DOI:** 10.3390/cells12091279

**Published:** 2023-04-28

**Authors:** Noureldien H. E. Darwish, Khaled A. Hussein, Khaled Elmasry, Ahmed S. Ibrahim, Julia Humble, Mohamed Moustafa, Fatma Awadalla, Mohamed Al-Shabrawey

**Affiliations:** 1Eye Research Center, Department of Foundational Medical Studies, Oakland University William Beaumont School of Medicine, Rochester, MI 48309, USA; 2Eye Research Institute, Oakland University, Rochester, MI 48309, USA; 3Clinical Pathology Department, Faculty of Medicine, Mansoura University, Mansoura 35111, Egypt; 4Oral and Dental Research Insitute, Department of Oral Medicine and Surgery, National Research Center, Cairo 11553, Egypt; 5Department of Oral Biology and Diagnostic Science, Dental College of Georgia, Augusta University, Augusta, GA 30912, USA; 6Department of Anatomy, Mansoura Faculty of Medicine, Mansoura University, Mansoura 35111, Egypt; 7Department of Ophthalmology, Visual and Anatomical Sciences, School of Medicine, Wayne State University, Detroit, MI 48201, USA; 8Department of Biochemistry, Faculty of Pharmacy, Mansoura University, Mansoura 35111, Egypt; 9Department of Pharmacology, School of Medicine, Wayne State University, Detroit, MI 48201, USA

**Keywords:** bone morphogenetic proteins, BMP4, Smad1/5/9, diabetic retinopathy, blood–retinal barrier function

## Abstract

Bone Morphogenetic Protein 4 (BMP4) is a secreted growth factor of the Transforming Growth Factor beta (TGFβ) superfamily. The goal of this study was to test whether BMP4 contributes to the pathogenesis of diabetic retinopathy (DR). Immunofluorescence of BMP4 and the vascular marker isolectin-B4 was conducted on retinal sections of diabetic and non-diabetic human and experimental mice. We used Akita mice as a model for type-1 diabetes. Proteins were extracted from the retina of postmortem human eyes and 6-month diabetic Akita mice and age-matched control. BMP4 levels were measured by Western blot (WB). Human retinal endothelial cells (HRECs) were used as an in vitro model. HRECs were treated with BMP4 (50 ng/mL) for 48 h. The levels of phospho-smad 1/5/9 and phospho-p38 were measured by WB. BMP4-treated and control HRECs were also immunostained with anti-Zo-1. We also used electric cell-substrate impedance sensing (ECIS) to calculate the transcellular electrical resistance (TER) under BMP4 treatment in the presence and absence of noggin (200 ng/mL), LDN193189 (200 nM), LDN212854 (200 nM) or inhibitors of vascular endothelial growth factor receptor 2 (VEGFR2; SU5416, 10 μM), p38 (SB202190, 10 μM), ERK (U0126, 10 μM) and ER stress (Phenylbutyric acid or PBA, 30 μmol/L). The impact of BMP4 on matrix metalloproteinases (MMP2 and MMP9) was also evaluated using specific ELISA kits. Immunofluorescence of human and mouse eyes showed increased BMP4 immunoreactivity, mainly localized in the retinal vessels of diabetic humans and mice compared to the control. Western blots of retinal proteins showed a significant increase in BMP4 expression in diabetic humans and mice compared to the control groups (*p* < 0.05). HRECs treated with BMP4 showed a marked increase in phospho-smad 1/5/9 (*p* = 0.039) and phospho-p38 (*p* = 0.013). Immunofluorescence of Zo-1 showed that BMP4-treated cells exhibited significant barrier disruption. ECIS also showed a marked decrease in TER of HRECs by BMP4 treatment compared to vehicle-treated HRECs (*p* < 0.001). Noggin, LDN193189, LDN212854, and inhibitors of p38 and VEGFR2 significantly mitigated the effects of BMP4 on the TER of HRECs. Our finding provides important insights regarding the role of BMP4 as a potential player in retinal endothelial cell dysfunction in diabetic retinopathy and could be a novel target to preserve the blood–retinal barrier during diabetes.

## 1. Introduction

Diabetic retinopathy (DR) is a microvascular complication of diabetes and a leading cause of legal blindness among the working-age population in the United States [1]. The cardinal pathological features of DR are the breakdown of the blood–retinal barrier (BRB) and retinal neovascularization [2]. Despite many recent advancements, therapeutic strategies to treat DR are still limited by their significant side effects and invasive approaches [3]. Thus, there is still an urgent need for new strategies to treat and prevent DR progression. A promising new target is a group of cytokines called bone morphogenic proteins (BMPs).

BMPs are growth factors and cytokines that belong to the transforming growth factor-β (TGF-β) superfamily. They have been known for their ability to induce bone and cartilage formation. There are several BMPs (BMP1-BMP15) and all of them except BMP1 have strong homology with TGF-β1 [4,5]. BMPs also play essential roles in embryonic development, physiological processes, and pathogenesis of various diseases [6]. Most BMPs exert their signaling effects through BMP receptor-I (BMPR-I) and BMPR-II. Upon activation by BMP, the BMPR-II kinase phosphorylates BMPR-I, which in turn activates the smad1/5/9 proteins to form a complex with smad-4. This complex translocates to the nucleus and modulates gene expression [7,8,9].

There are four BMPR1s, Alk1, Alk2, Alk3 (Bmpr1a), and Alk6 (Bmpr1b), that have been characterized [10,11,12]. BMP2, BMP4, and BMP6 bind with a high affinity to ALK2, ALK3, and ALK6 [12,13]. This interaction activates signaling pathways that are implicated in endothelial cell dysfunction and associated with various vascular diseases, including atherosclerosis and coronary heart disease [14,15]. However, the role of BMP signaling in DR is still to be clarified. Our previous studies support the role of BMP2 in retinal endothelial cell dysfunction in DR via activation of both smad (canonical) and p38 MAPK (non-canonical) [12]. The role of BMP4 in endothelial cell dysfunction has also been demonstrated in various experimental models [15,16,17,18]. Recent studies demonstrated that platelet BMP4 is implicated in vascular inflammation and remodeling following wire-induced injury [18,19]. However, the role of BMP4 in retinal endothelial cell function and DR has not yet been characterized.

The goal of this study was first to characterize the impact of diabetes on retinal levels of BMP4 and second to investigate the direct effect of BMP4 on retinal endothelial cell barrier function.

## 2. Materials and Methods

### 2.1. Postmortem Human Retina Samples

We obtained postmortem human eyes of non-diabetic and diabetic retinopathy subjects from Georgia Eye Bank (Atlanta, GA, USA) and Eversight (Ann Arbor, MI, USA) (40–60 years old, *n* = 4–6 per group). We dissected the retinas out of the eyeball and homogenized them, and the total protein was used for Western blot studies. Some other eyeballs were embedded in paraffin and were cut by microtome (20 μm thick sections) for immunolocalization studies.

### 2.2. Experimental Mice

Animal studies were carried out at Eye Research Institute, Oakland University (Rochester, MI, USA) animal facility. The animal protocols were approved by the Institutional Animal Care and Use Committee (IACUC) (protocol number 2022-1159). Akita mice were used as a murine model for type-1 diabetes mellitus (DM). Wild-type (C57BL/6) and Akita mice were purchased from Jackson Laboratories (Bar Harbor, ME) (*n* = 6–7 per group). All animals were group-housed for 6 months, and exposed to the standard 12 h light/12 h dark cycle. The mice were provided with food and water ad libitum and kept at a temperature range of 22–24 °C.

### 2.3. Human Retinal Endothelial Cells

Human retinal endothelial cells (HRECs) (Cell Systems, Kirkland, WA, USA) were grown using Endothelial Basal Medium-2 (Cell Systems).

### 2.4. Assessment of BMP4 Expression and Localization in Human and Mouse Retinas

We explored the changes in BMP4 protein expression and localization in the postmortem human retina of diabetic retinopathy compared to non-diabetic subjects using immunofluorescence and Western blot techniques. For immunofluorescent staining, eyeballs were embedded in paraffin and then sectioned at 8 μm. The sections were de-paraffinized and incubated with anti-BMP4 primary antibody (Thermo Scientific, Waltham, MA, USA) at a concentration of 1:100 and the blood vessel marker Griffonia Simplicifolia Biotinylated Isolectin-B4 (Vector Lab, Newark, CA, USA) at a concentration of 15 μg/mL at 4 °C overnight. The sections were then washed with PBS–Triton X-100 two–three times. Then, the sections were incubated with the proper secondary antibodies, Oregon Green labeled to visualize the anti-BMP4 (1:500) and Texas Red Conjugated Avidin (7.5 μg/mL) (Vector Lab, Newark, CA, USA) to visualize the isolectin-B4 Invitrogen, Eugene, OR, USA), and cover-slipped with Fluoro-shield containing DAPI (Sigma-Aldrich, St. Louis, MO, USA) to label the nuclei. An Axioplan-2 fluorescent microscope (Carl Zeiss, Göttingen, Germany) with a high-resolution microscope (HRM) camera was used to capture images using the Zeiss Axiovision digital image processing software (version 4.8). A similar staining procedure was performed on mouse retinal cryosections. We first prepared retinal sections by embedding fresh mouse eyeballs inside the tissue. Tek Optical Cutting Temperature or O.C.T. compound (Sakura Finetek, Torrance, CA, USA) followed by cutting 12 μm thick retinal sections by a cryostat. Retinal sections were fixed in 4% for 10 min paraldehyde before.

Immunostaining of retinal sections from human and mice were used to characterize the expression level and localization of BMP4 in relation to retinal vessels. For this purpose, we used isolectin-B4 as a vascular marker and specific antibody against BMP4. To obtain more representative results, we examined 3 fields of each retinal section from at least 3 human subjects or mice of each group (diabetic versus non-diabetic). Then, the immunoreaction was evaluated after adjusting the autofluorescence/background to capture the most specific immunoreaction using the same setting for all groups. 

Western blot analysis was used to confirm the BMP4 expression in the postmortem human retinas and retinas of experimental mice. Retinas were lysed in RIPA buffer supplemented proteinase/phosphatase inhibitor cocktail (1:100 (*v*/*v*) (Thermo Scientific, Waltham, MA, USA)). Then, the tissue homogenates were centrifuged at 12,000× *g* at 4 °C for 30 min. Protein concentration was determined by BCA Protein Assay (Thermo Scientific, Waltham, MA, USA). An equal protein amount was boiled in a Laemmli sample buffer. The samples were loaded onto sodium dodecyl sulfate-polyacrylamide gels (SDS-PAGE) to run the gel electrophoresis. Proteins were then transferred to nitrocellulose membranes, which were blocked with 5% milk in 1X TBST. Membranes were then incubated with the following primary antibodies: BMP4 (Abcam) and β-actin (Cell signaling, Danvers, MA, USA). Blots were incubated with the proper peroxidase-conjugated secondary antibody and visualized with the enhanced chemiluminescence (ECL) Western blot detection system (Thermo Scientific). The optical density of the bands was assessed using ImageJ software.

### 2.5. In Vitro Studies

#### 2.5.1. Assessment of ZO-1 Distribution in HRECs Cells Using Immunofluorescence Staining

Human retinal endothelial cells (HRECs) were grown using Endothelial Basal Medium-2 (Cell Systems, Kirkland, WA, USA). HRECs (70–80% confluent) were treated with rhBMP4 (50 ng/mL) for 48 h, then stained with ZO-1 antibodies according to our previous studies [12,20,21]. HRECs were fixed using 4% paraformaldehyde, followed by a blockage in normal goat serum. Thereafter, HRECs were incubated with antibodies against ZO-1 (Thermo Scientific) at a concentration of 1:100 overnight at 4 °C followed by incubation with the appropriate secondary antibodies Alexafluor (Invitrogen, Eugene, OR, USA) at a concentration of 1:500. Finally, nuclei were stained with 4′,6 diamidino-2-phenylindole (DAPI) mounting medium (Vector Laboratories, Burlingame, CA, USA), and images were taken with confocal microscopy (LSM 510; Carl Zeiss, Thornwood, NY, USA).

#### 2.5.2. Assessment of Human Retinal Endothelial Cell Barrier Function

The vascular endothelial growth factor (VEGF) and bone morphogenetic proteins (BMPs) are key regulators for the BRB. several studies have shown that BMP4 mediate its effects through VEGF-A/VEGFR2 and angiopoietin-1/TIE2 signaling stimulation [22,23]. On the other hand, the BMP signaling was regulated by different regulating factors such as noggin, chordin, and gremlin [24]. 

To study the changes in HRECs barrier function under various treatments, HRECs were grown in 96-well electrode arrays (catalog # 96W20idf PET, Applied Biophysics Inc., Troy, NY, USA) coated with 100 µM cysteine and 0.02% gelatin. After growing to confluence, cells were serum starved for 8 h. Cells were then treated with rhBMP4 (50 ng/mL: R&D Systems, Minneapolis, MN, USA) with or without the following inhibitors of BMP signaling: Noggin, (200 ng/mL; R&D Systems), LDN-193189 (LDN1, 200 nM; Sigma-Aldrich, St Louis, MO, USA), and LDN-212854 (LDN2, 200 nM; Sigma-Aldrich). The BMP was prepared by dissolving in 4 mM HCl (according to the manufacturer) and then diluted with the media to obtain our final concentration. The choice of BMP4 (50 ng/mL) for 48 h was based on our previous experience in which we had a consistent and optimal effect at this concentration [12,20,21]. LDN1 is a selective inhibitor of ALK2 and ALK3 with higher affinity to ALK3, while LDN2 is an inhibitor of ALK2 with weaker effects on ALK1 and ALK3 [25,26]. HRECs were also subjected to BMP4-treatment with or without inhibitors of VEGFR2 (SU5416, 10 μM: Sigma-Aldrich), endoplasmic reticulum (ER) stress inhibitor Phenylbutyric acid (PBA, 30 μmol/L; Sigma-Aldrich, St. Louis, MO, USA), p38 (SB202190, 10 μM; R&D Systems, Minneapolis, MN, USA) or ERK (U0126, 10 µM; R&D Systems). The changes in HRECs barrier function were assessed by measuring normalized transcellular electrical resistance (TER) using the electric cell-substrate impedance sensing [ECIS^®^Zθ (theta)] instrument (Applied Biophysics Inc.) as previously described [27,28,29]. On the other hand, ECIS provides real-time information on cell capacitance that reflects cell survival and adherence throughout the entire period of the experiment [30].

#### 2.5.3. Assessment of the Effect of BMP4 Treatment on Phospho-Smad 1/5/9 and Phospho-p38 Levels

Western blot analysis was used to assess the effect of BMP4 on the canonical and non-canonical pathways in HRECs, using phospho-smad 1/5/9 as a marker for the canonical pathway and p38 MAPK/phospho-p38 MAPK for the non-canonical pathway. After treatment with rhBMP4 (50 ng/mL) for 30 min, HRECS were collected and the nuclear extract was prepared using the nuclear extraction kit (Abcam INC, Cambridge, MA, USA). Other HREC groups were subjected to rhBMP4 treatment for 24 h. Using the Western blot protocol mentioned previously, blotted proteins, along with loading control histone deacetylase (HDAC) (Abcam Inc., Cambridge, MA, USA) were incubated with anti-phospho-smad 1/5/9 (Abcam Inc., Cambridge, MA, USA), p38 MAPK/phospho-p38 MAPK (Thermo Scientific, San Diego, CA, USA), and β-actin (Cell signaling, Boston, MA, USA).

#### 2.5.4. Measurement of Matrix Metalloproteinase Activity (MMPs) Activities

We measured the activity of matrix metalloproteinase-2 and 9 (MMP2 and MMP9) in HRECs protein extracts by using a Fluorometric SensoLyte 520 Generic MMP Assay Kits (AnaSpec, Fremont, CA, USA) according to the manufacturer’s instructions. The kit uses a 5-FAM/QXL™520 fluorescence resonance energy transfer (FRET) peptide linked to different MMP substrates where the fluorescence of 5-FAM is quenched by QXL™520. The activities of MMP were measured by estimating the fluorescence intensity of 5-FAM released at the cleavage of MMPs at excitation/emission wavelengths = 490 nm/520 nm. 

#### 2.5.5. Statistical Analysis

GraphPad Prism 8 was used to conduct statistical analyses. The differences between experimental groups were assessed using the two-tailed *t*-test or one-way analysis of variance (ANOVA) followed by the Tukey test. Results are presented as means ± SD. A *p*-value < 0.05 was considered statistically significant. 

## 3. Results

### 3.1. Diabetes Upregulates Retinal BMP4 in Human Subjects and Experimental Mice

Western blot analysis of the retinal homogenate of postmortem human eyes showed a significant increase in BMP4 expression in diabetic subjects compared to non-diabetic subjects (~3.5-fold, *p* = 0.0003). The high expression of BMP4 in diabetic retinas was confirmed with the immunofluorescence staining, which showed a marked increase in BMP4 immunoreactivity in retinal vessels and photoreceptors (Figure 1). Similarly, we also observed a significant increase in retinal levels of BMP4 in 6-month-old diabetic mice (Akita) compared to the control group (~1.7-fold, *p* = 0.027). Immunolocalization of BMP4 in retinal sections of control and diabetic mice showed a remarkable increase in BMP4 immunoreactivity in the retinal vessels of diabetic mice compared to the control (Figure 2).

### 3.2. BMP4 Disrupts Human Retinal Endothelial Barrier Function In Vitro

We examined the effect of BMP4 (50 ng/mL) on the barrier function of HRECs in vitro by assessment of the changes in the tight junction proteins (ZO-1) by immunofluorescence and transcellular electrical resistance (TER) by ECIS. In control cells, ZO-1 formed a cellular border pattern, while in BMP4-treated cells, ZO-1 distribution appeared markedly affected, demonstrating that BMP4 disrupts ZO-1 organization in HRECs (Figure 3). ECIS studied for over 200 h also showed a significant reduction of TER by BMP4 compared to vehicle-treated HRECs (Figure 4). The effect of BMP4 on TER of HRECs was reversed by inhibitors of BMP/ALK signaling such as noggin, which inhibits binding of BMP4 to its receptors, and LDN1 and LDN2, which inhibit ALK2 and ALK3 (*p* < 0.001) (Figure 4 and Figure 5). Moreover, we investigated various molecular targets that may play a role in BMP4-induced endothelial barrier disruption. Thus, we extended our ECIS studies by treating HRECs with BMP4 in the presence or absence of pharmacological inhibitors such as VEGFR2 inhibitor (SU5416), p38 MAPK inhibitor (SB202190), ERK inhibitor (U0126), and ER stress inhibitor (phenyl butyric acid or PBA). There was a partial but significant restoration of TER in BMP4-treated HRECs in the presence of VEGFR2 inhibitor (*p* = 0.038) and p38 pathway inhibitor (*p* = 0.005) (Figure 6 and Figure 7). On the other hand, the ERK inhibitor further reduced TER, thereby enhancing the permeability effect of BMP4. The other tested pathway inhibitor of ER stress, PBA, did not show significant changes in TER compared to HRECs treated only with BMP4.

### 3.3. BMP4 Activates the Canonical Smad Pathway and Non-Canonical p38-MAPK Pathway in HRECs

The smad effectors (smad1/5/9 and smad4) represent the main players of the canonical pathway of the BMP4 signaling. Here, we evaluated the direct effect of BMP4 on the total and nuclear levels of phosphorylated (activated) smad 1/5/9 (Figure 8A,B) in HRECs by Western blot. Treatment of HRECs with BMP4 induced a remarkable increase in the total level of nuclear translocation of p-smad1/5/9 compared to the vehicle-treated group (~1.5-fold, *p* = 0.039).

Inhibition of p38 MAPK reversed the pro-permeability effect of BMP4, showing the importance of the non-canonical pathway. To explore this further, we tested the direct effect of BMP4 on p38 activation in HRECs. Western blot analysis showed a significant increase in phospho-p38 expression in BMP4-treated cells compared to vehicle-treated cells (*p* = 0.0138) and a (1.5 ± 0.5) fold increase (Figure 8C).

### 3.4. BMP4 Increases the Activity of Matrix Metalloproteinases (MMPs) in HRECs

Matrix metalloproteinases (MMPs) are key players in retinal endothelial cell dysfunction in diabetic retinopathy. MMPs contribute not only through basement membrane degradation, but also by allowing endothelial cells to detach and migrate into new tissue, and releasing ECM-bound proangiogenic factors such as VEGF and TGFβ [31].

Treatment of HRECs with rhBMP4 caused significant increases in the activity of both MMP-2 (~4-fold vs. control, *p* = 0.0022) and MMP-9 (~8-fold vs. control, *p* < 0.0001), suggesting MMPs as potential targets to the activated BMP4 signaling in HRECs (Figure 9). 

## 4. Discussion

The role of BMP signaling in endothelial cell dysfunction has attracted a lot of researchers who showed the pro-permeability and pro-angiogenic potential of the activated BMP signaling [12,22,32,33,34]. However, the association between BMP4 signaling and endothelial cell dysfunction in diabetic retinopathy has not yet been elucidated. Here, we provide the first evidence for the increased levels of BMP4 in the retinas of diabetic humans and mice (Akita) compared to their control groups. Moreover, we established the pro-permeability effect of BMP4 on human retinal endothelial cells. Consistent with our results, the Dong group has reported that BMP4 and SMAD9 were highly expressed in STZ-induced diabetic rats (~2-fold) in comparison to the control group [35].

The downstream analysis established that BMP4 regulates the expression of the different survival factors through its binding to BMPR1s, particularly ALK2, ALK3, and ALK6 [36,37]. BMP4 mediates its biological functions through the activation of two independent pathways, the smad (canonical) and p38 MAPK (non-canonical) pathways [36,37,38]. On the other hand, the expression of *BMPR1* is further increased upon BMP4 exposure, which can explain the strong correlation between BMP4 and BMPR1 [38]. In our study, we showed a significant increase in BMP4 retinal expression in diabetic human subjects and experimental mice compared to the non-diabetic groups. Moreover, BMP4 was primarily localized in retinal vessels. Our data also demonstrated a disruptive effect of BMP4 on the barrier function of HRECs, as demonstrated by the significant decrease in TER in BMP4-treated cells and the disruption of ZO1 localization in HRECs. This disruption was prevented by pharmacological inhibitors of BMP/ALK signaling, including noggin, LDN1, and LDN2. Altogether, our data highlight the potential role of the activated BMP4 signaling in regulating retinal endothelial cell dysfunction during diabetes. BMP signaling is essential for the healthy development of the retinal vasculature. However, diseases such as diabetes are associated with an increase in BMP expression. Thus, when BMP4 levels are elevated beyond their normal levels, BMP4 signaling leads to the breakdown of the blood–retinal barrier [22,39].

To understand the underlying mechanism of BMP4-induced retinal endothelial cell permeability, we investigated its impact on the levels of phspho-smad1/5/9 and its nuclear translocation, which is required to induce the transcription activity of BMP4-dependent genes in HRECs. Our Western blot analysis showed a marked increase in the levels of p-smad1/5/9 and its translocation to the nucleus in BMP4-treated cells suggesting the importance of smad-dependent signaling in inducing the effect of BMP4 on HRECs. In addition to the smad system, activation of the non-canonical pathway, as represented by p38 MAPK, has also been shown to be involved in the biological effects of BMP4 [22,40]. Our experiments also demonstrated a significant increase in the levels of phosphorylated-p38 MAPK in HRECs by BMP4. Next, we tested the ability of different inhibitors, including inhibitors of VEGFR2, p38 MAPK, ERK, and ER stress, to preserve the barrier function of BMP4-treated HRECs. We noticed significant restoration of the barrier function in BMP4-treated HRECs by VEGFR2 and p38 pathway inhibitors compared to the other inhibitors. 

Our results suggest that BMP signaling via VEGFR2 and p38 MAPK could be involved in BMP4-induced endothelial cell dysfunction in DR. Another recent study has reported the role of BMP signaling via p38 MAPK in promoting tumor angiogenesis [41]. In addition, many studies have reported the cross-talk between BMP4 and VEGF. VEGF has been shown to induce BMP expression in the microvascular endothelial cells, including BMP2 and BMP4 [42,43]. VEGF has been recognized as a downstream target from smad and p38 MAPK [33,44]. VEGF and BMP4 have been shown to enhance the high expression of PECAM1 and VE-cadherin, which play an important role in endothelial cell permeability and the development of new blood vessels [34,42,45]. BMP signaling plays an essential role in blood vessel formation in the retina and throughout the body [12,46,47]. Taken together, our data demonstrate the importance of BMP4 signaling in endothelial cell dysfunction through both the canonical and non-canonical pathways.

Regarding the retinal–vascular microenvironment, We and others have already established the permeability changes in diabetic mice in comparison to control using BSA immunostaining or Evans blue permeability assays compared with that in the normal controls [28,48,49,50]. The extracellular matrix (ECM) is known to play a supporting role for different tissues and also contributes to different functions, including regulation of the cell cycle and cell motility, and apoptosis. The ECM is made up of several molecules, including collagen, elastin, and adhesion proteins, such as fibronectin and laminin; as well as proteases called matrix metalloproteases (MMPs) [51]. The MMPs belong to the endopeptidases family that contains contain zinc and can degrade and remodel the ECM’s proteins [52].

The MMPs family consists of 23 members which are divided based on their sub-cellular distribution and specificity for components of the ECM. The MMPs are divided into membrane-type matrix metalloproteases (MT-MMPs), collagenases, stromelysins, matrilysins, and gelatinases members (MMP-2 and MMP-9) [31]. Activation of retina MMPs, particularly MMP-2 and MMP-9, is an early event, that contributes to mitochondrial injury and promotes retinal apoptosis of vascular cells, including pericytes and endothelial cells [53]. Different mechanisms regulate MMPs in the retina, including tissue inhibitors of MMPs (TIMPs), cleavage of MMPs zymogens, and epigenetic modifications. Thus, inhibitors of MMP2 and MMP9 have been suggested to mitigate capillary cell apoptosis and the progression to the angiogenic stage of DR [54]. Several studies have demonstrated the effects of hyperglycemia on MMP regulation in vascular cells [55,56]. A recent case-control study has reported a significant increase in average MMP9 values in the vitreous of diabetic groups compared to the non-diabetic group [56]. It was observed that there is a direct correlation between the MMP-9 levels and the DR duration. In addition, it has been reported that the risk of DR incidence increased with high levels of MMP-9 [56]. Consistent with the potential role of BMP4 in DR, our data demonstrated significant increases in the activity of MMP2 and MMP9 in HRECs by BMP4 treatment. These data underscore MMP2 and MMP9 as downstream mediators from retinal BMP4 in DR and a potential underlying mechanism by which BMP4 modulates retinal endothelial barrier function in DR.

## 5. Conclusions

The current study reports upregulation of BMP4 in retinal vessels of diabetic human subjects and experimental mice and significant disruption of human retinal endothelial cell barrier function by BMP4 treatment. BMP4 activated smad1/5/9 and p38 MAPK, as well as MMP2 and MMP9, suggesting that the role of BMP4 in retinal endothelial cell dysfunction could be via activating both the canonical and non-canonical pathways. Altogether, our data propose BMP4 as a novel player that contributes to microvascular dysfunction in DR and could be a new therapeutic target, inhibition of which will mitigate the permeability effect of diabetes in DR.

## Figures and Tables

**Figure 1 cells-12-01279-f001:**
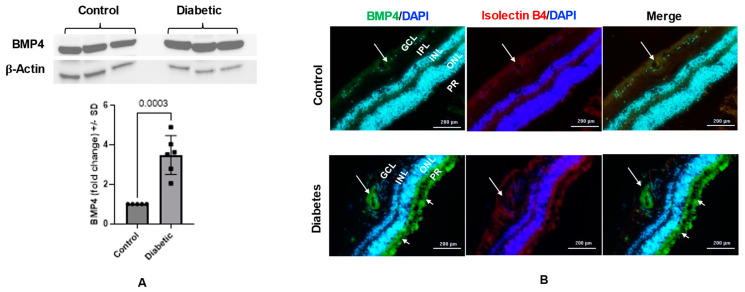
BMP4 expression in postmortem human retinas (diabetic versus non-diabetic). (**A**) Western blot analysis of the BMP4 expression in the retinas of diabetic patients versus non-diabetic subjects. Densitometry analysis shows a significant increase in the levels of BMP4 in the retinas of diabetic subjects compared to non-diabetic group (~3.5-fold, *p* = 0.0003). Results are presented as fold ± SD; *n* = 5–6 per group. Unpaired *t*-test was used to compare the BMP4 levels in diabetic retinas versus control. (**B**) Immunofluorescence of BMP4 (green) and isolectin-B4 (red) co-localization suggesting increased BMP4 expression in retinal vessels (long arrows) in diabetic subjects compared to the control. Furthermore, BMP4 immunoreactivity increased in photoreceptor layer (short arrows) of diabetic subjects. *n* = 4–5. GCL = ganglion cell layer, IPL = inner plexiform layer, INL = inner nuclear layer, ONL = outer nuclear layer, PR = photoreceptors.

**Figure 2 cells-12-01279-f002:**
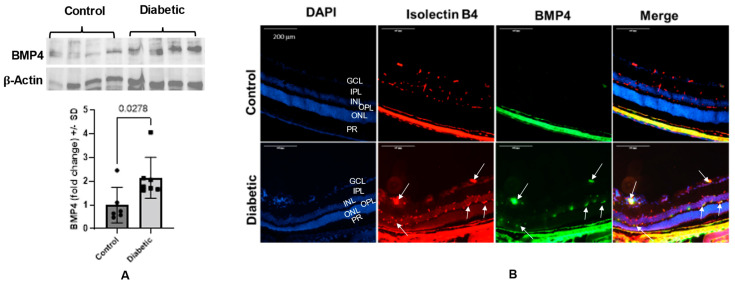
BMP4 expression in mouse retina. (**A**) Western blot analysis of the BMP4 expression in the retinas of 6-month diabetic Akita mice versus age matching control. Densitometry analysis shows significant increase in the levels of BMP4 in the retinas of Akita mice compared to control (~1.7-fold, *p* = 0.027). Results are presented as fold ± SD; *n* = 6–7 per group. Unpaired *t*-test was used to compare the BMP4 levels in Akita retinas versus control. (**B**) Immunolocalization of BMP4 in retinal sections of control and diabetic mice shows a remarkable increase in BMP4 (green) immunoreactivity in retinal vessels (red) of diabetic mice compared to the control. Note the colocalization of BMP4 with the vascular marker (arrows), *n* = 6–7. GCL = ganglion cell layer, IPL = inner plexiform layer, INL = inner nuclear layer, ONL = outer nuclear layer, OPL = outer plexiform layer, PR = photoreceptors.

**Figure 3 cells-12-01279-f003:**
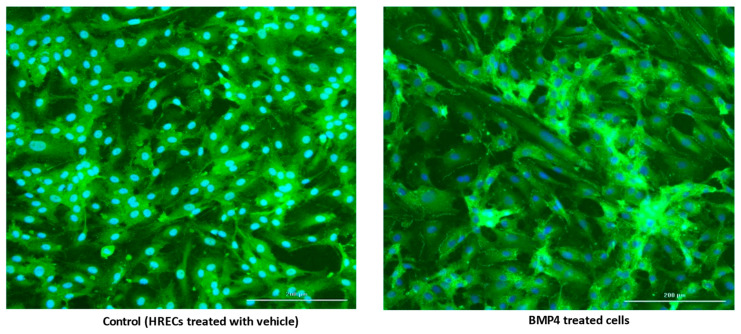
Effect of BMP4 on ZO-1. Representative photographs of ZO-1 immunofluorescence (green) and nuclear stain (DAPI-Blue) in HRECs cells treated with BMP4 (50 ng/mL) versus vehicle-treated controls. ZO-1 distribution appears to be disrupted in BMP4-treated HRECs in comparison to the normal distribution pattern in vehicle-treated HRECs; *n* = 4/group; scale bar = 200 µm.

**Figure 4 cells-12-01279-f004:**
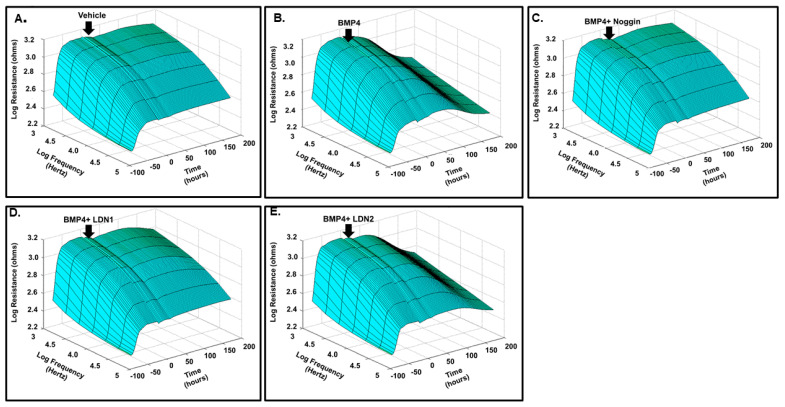
Three-dimensional model representation for the effect of BMP4 on HRECs barrier function. Thereafter, monolayers of HRECs were treated with BMP4 (50 ng/mL) in the presence or absence of noggin, LDN1, or LDN2. The barrier integrity (trans-endothelial electrical resistance or TER) was monitored over a 200 h period (*z*-axis), across various frequencies represented as log values on the *x*-axis. As shown in (**B**) BMP4-treatment reduced the TER compared to the vehicle (**A**). Noggin, LDN1, and LDN2 (**C**–**E**) reversed the effect of BMP4 on the TER of HRECs.

**Figure 5 cells-12-01279-f005:**
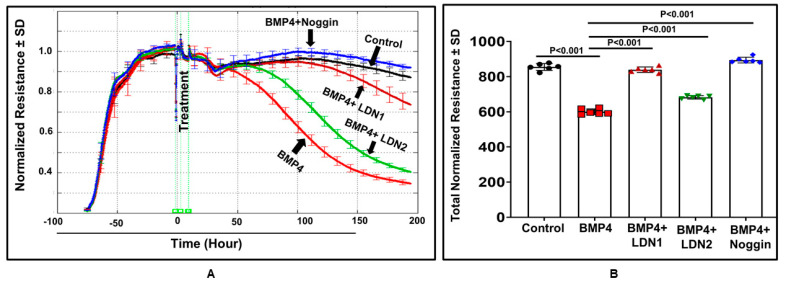
Normalized resistances of BMP4-treated HRECs measured at a frequency of 4000 Hz by ECIS (**A**), and the statistical analysis (**B**). The confluent mature monolayers of HRECs have been confirmed by the plateau phase of electric resistance, where the resistance reached 1000 ohms. to observe the changes in the resistance in real time, we present the data as normalized TER resistance calculated by dividing the resistance of each well measured in ohms at each time point by the baseline resistance (ohms) obtained before the adding of the BMP4 and plotted as a function of time. There is a significant drop in the resistances of BMP4-treated (50 ng/mL) HRECs over the experimental period, compared to vehicle-treated control HRECs. Results are demonstrated as mean ± SD, *n* = 4−6/group; *p* < 0.001 versus vehicle-treated control HRECs. Black line and circles: control; red line and rectangular: BMP4 (50 ng/mL); dark-red line and triangles: BMP4 (50 ng/mL) + LDN1 (200 nM); green line and triangles: BMP4 (50 ng/mL) + LDN2 (200 Nm); and blue line and diamonds: BMP4 (50 ng/mL) + Noggin (200 ng/mL). As shown in the figure, BMP treatment led to a significant wereduction in HRECs barrier integrity, while BMP4 inhibitors preserved barrier function. Results are presented as mean ± SD.

**Figure 6 cells-12-01279-f006:**
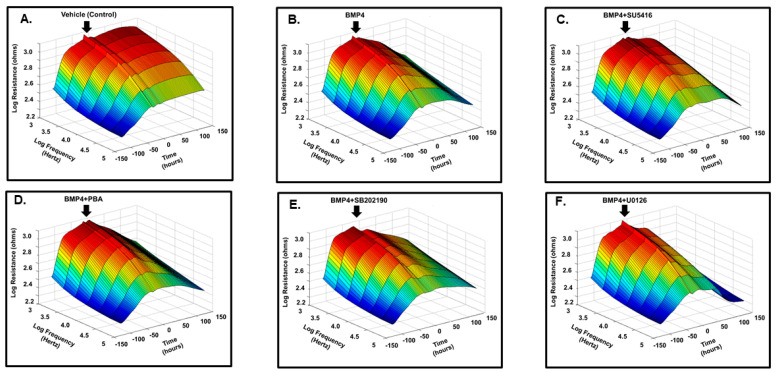
Three-dimensional model demonstration for the effect of BMP4 treatment on HRECs barrier function. The changes in HRECs barrier function after BMP4 treatment for 1–5 days with or without inhibitors of VEGFR2 (SU5416, 10 μM), endoplasmic reticulum (ER) stress inhibitor phenylbutyric acid (PBA, 30 μmol/L), p38 (SB202190) or ERK (U0126, 10 µM). BMP4-treatment was initiated after HRECs formed a confluent monolayer indicated by the plateau in the resistance (*y*-axis). The barrier integrity was monitored over 200 h period (represented on the *z*-axis) and across various frequencies represented as log values on the *x*-axis. As shown in (**B**) BMP4 treatment reduced the TER compared to the vehicle (**A**). VEGFR2 inhibitor (SU5416) and p38 inhibitor (SB202190) (**C**,**E**) partially reversed the effect of BMP4 on the TER of HRECs. ER stress inhibitor (PBA) and ERK (U0126) (**D**,**F**) have the least effect.

**Figure 7 cells-12-01279-f007:**
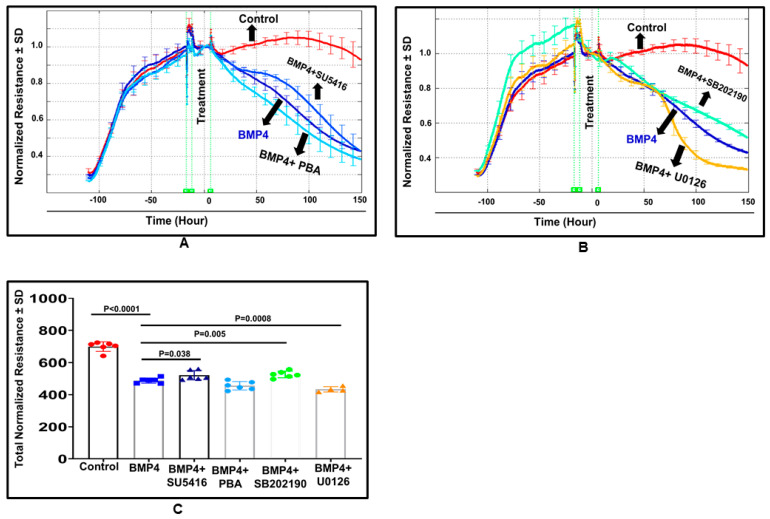
Normalized resistances of BMP4-treated HRECs estimated at a frequency of 4000 Hz by ECIS (**A**,**B**), and the statistical analysis (**C**). There was a significant decline in the resistances of HRECs with direct BMP4-treatment (50 ng/mL) over the experimental period in comparison to vehicle-treated control HRECs. There was a partial but significant restoration of TER in BMP4-treated HRECs by VEGFR2 inhibitor (SU5416) (between 50–100 h interval) and p38 pathway inhibitor (SB202190) (between 75–150 h interval) (*p* = 0.038, *p* = 0.005, respectively) compared to BMP4-treated cells. ERK inhibitor (U0126), and ER stress inhibitor (phenyl butyric acid or PBA) did not show any effects on TER. Results are presented as mean ± SD, *n* = 4–6/group; *p* < 0.01. Control: red line; BMP4 (50 ng/mL): dark-blue line; BMP4 (50 ng/mL) + VEGFR2 inhibitor (SU5416, 10 μM): light-blue line; BMP4 (50 ng/mL) + PBA (30 μmol/L): sky-blue line; BMP4 (50 ng/mL) + p38 pathway inhibitor (SB202190, 10 μM): green; and BMP4 (50 ng/mL) + ERK inhibitor (U0126, 10 μM): orange.

**Figure 8 cells-12-01279-f008:**
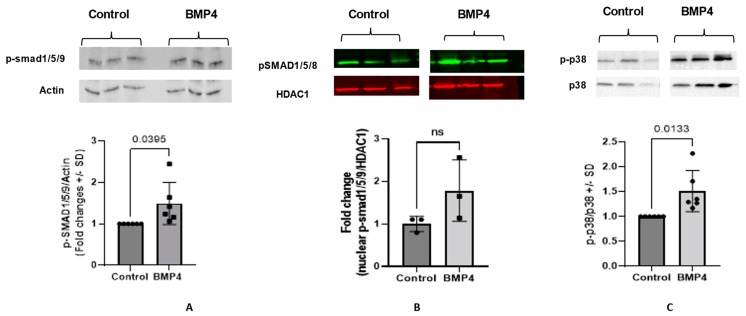
Western blot analysis of the effect of BMP4 on p-smad 1/5/9 and p38. Densitometry analysis shows significant increase in the levels of total p-smad1/5/9 in human retinal endothelial cells (HRECs) compared to the control (**A**). Nuclear levels of p-smad show a modest non-significant increase (**B**). There was a significant increase in the levels of p-p38 in human retinal endothelial cells (HRECs) compared to the control (**C**). Unpaired *t*-test was used to compare the effect of BMP4 on p-p38 and p-smad1/5/9 compared to control. *n* = 3–6 total.

**Figure 9 cells-12-01279-f009:**
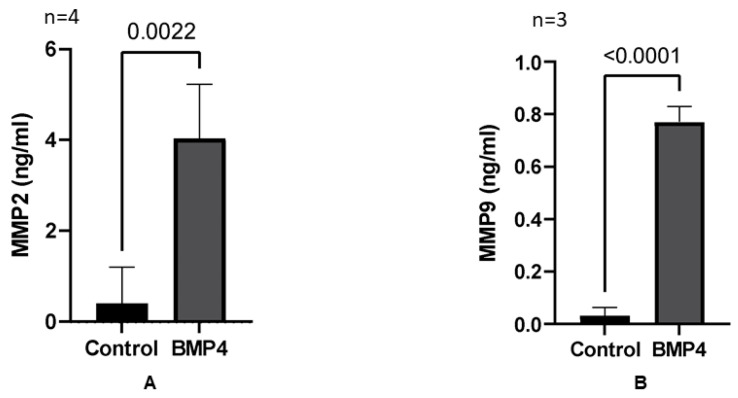
Matrix metalloproteinase (MMP) activity after BMP4-treatment (Sensolyte assay). (**A**) MMP-2 activity was significantly increased after exposure to BMP4 (~4-fold, *p* = 0.0022). (**B**) MMP-9 activity was significantly increased after exposure to BMP4 (~8-fold, *p* < 0.0001). The results are presented as mean ± SD; *n* = 3–4 per group.

## Data Availability

The original contributions presented in the study are included in the article. Further inquiries can be directed to the corresponding author.

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
