# Peer review of "Bone Morphogenetic Protein-4 Impairs Retinal Endothelial Cell Barrier, a Potential Role in Diabetic Retinopathy"

_cells, 2023, doi:10.3390/cells12091279_

Round 1
Reviewer 1 Report
This manuscript (MS) by Darwish et al. aims to demonstrate bone morphogenetic protein-4 (BMP-4) and its associated signaling molecules in regulating endothelial tight junctions. They demonstrated that BMP-4 was upregulated in the retinas from diabetic patients and Akita mice compared to the controls. They further used cultured human retinal endothelial cells (RECs) and pharmacological inhibitors to show that BMP-4 elicited decreases of transcellular electrical resistance (TER) were blocked by noggin, LDN1, LDN2, as well as inhibitors of VEGFR2 and p38. There are a few issues that require the authors’ attention.
1. In the Materials and Methods, please describe the conditions of how the immunofluorescent images were taken, and whether there was any contrast/image enhancement. It is unclear why the DAPI images appear to have two different colors (Figure 1B; Figure 3).
2. In the Materials and Methods, please indicate that data are presented as Mean +/- SD. Otherwise, please denote it in the figure legend of "Figure 8."
3. Figure 2A, please correct the labels of the Western blots. Make sure it follows the same format as in Figure 1A.
4. Most of the Western blot images are not good or are inconsistent. For example, the beta-actin in Figure 2A appears to have double bands but not in Figure 1A; BMP4 bands in Figure 2A are rather messy compared to Figure 1A. The quality of the blots in Figure 8A are not good. In Figure 8B, the loading control (HDAC1) and pSMAD1/5/8 do not appear to be the same blot/gel but two independent trials. The blots in Figure 8C have similar problems as Figure 8B.
5. Figure 3 has epifluorescence images of ZO-1 of cultured RECs treated with BMP4 or vehicle. BMP4 treated RECs do show higher ZO-1 staining. Since these are epifluorescence images, and if the detergent Triton-X 100 was used, it would be impossible to differentiate whether the staining is cytosolic or membrane-bound.
6. In pg. 10, ln 309-310, the authors mentioned that MMPs are key players in REC “dysfunction in diabetic retinopathy.” Please elaborate on the roles of MMPs beyond dysfunction.
7. The Discussion section could be more in depth.
Author Response
This manuscript (MS) by Darwish et al. aims to demonstrate bone morphogenetic protein-4 (BMP-4) and its associated signaling molecules in regulating endothelial tight junctions. They demonstrated that BMP-4 was upregulated in the retinas from diabetic patients and Akita mice compared to the controls. They further used cultured human retinal endothelial cells (RECs) and pharmacological inhibitors to show that BMP-4 elicited decreases of transcellular electrical resistance (TER) were blocked by noggin, LDN1, LDN2, as well as inhibitors of VEGFR2 and p38. There are a few issues that require the authors’ attention.
- In the Materials and Methods, please describe the conditions of how the immunofluorescent images were taken, and whether there was any contrast/image enhancement. It is unclear why the DAPI images appear to have two different colors (Figure 1B; Figure 3).
Response
- Thanks for your suggestion; in revision, the materials and methods section has been updated. Lines 122 and 128 read ``Immunostaining of retinal sections from human and mice were used to characterize the expression level and localization of BMP4 in relation to retinal vessels. For this purpose, we used isolectin-B4 as vascular marker and specific antibody against BMP4. To get more representative results, we examined 3 fields of each retinal section from at least 3 human subjects or mice of each group (diabetic versus non-diabetic). Then, the immunoreaction was evaluated after adjusting the autofluorescence/background to capture the most specific immunoreaction using the same setting for all groups``.
- The difference in blue color between the human section and HRECs could be attributed to the fact that human retinal sections are paraffin sections which cause extensive autofluorescence and nonspecific background reaction.
- In the Materials and Methods, please indicate that data are presented as Mean +/- SD. Otherwise, please denote it in the figure legend of "Figure 8."
Response
- Thanks for your suggestion; in revision, the materials and methods section (Statistical Analysis) has been updated. Line 206 reads `` Results are presented as means ± SD ``.
- Figure 2A, please correct the labels of the Western blots. Make sure it follows the same format as in Figure 1A.
Response
- Thanks for your suggestion; in revision, figure 2A has been updated.
- Most of the Western blot images are not good or are inconsistent. For example, the beta-actin in Figure 2A appears to have double bands but not in Figure 1A; BMP4 bands in Figure 2A are rather messy compared to Figure 1A. The quality of the blots in Figure 8A are not good. In Figure 8B, the loading control (HDAC1) and pSMAD1/5/8 do not appear to be the same blot/gel but two independent trials. The blots in Figure 8C have similar problems as Figure 8B.
Response
For all Western blot images, we have already shared original blots with the editor with the original submission. Furthermore, in revision; for Figure 8, the updated images have been added. Both HDAC1 and p-smad1/5/9 are the same blot but HDAC1 picture was relatively stretched (please see the original blot below
- Figure 3 has epifluorescence images of ZO-1 of cultured RECs treated with BMP4 or vehicle. BMP4 treated RECs do show higher ZO-1 staining. Since these are epifluorescence images, and if the detergent Triton-X 100 was used, it would be impossible to differentiate whether the staining is cytosolic or membrane-bound.
Response
- We agree with you. In the revision, Lines 241 and 243 read ``In control cells, ZO-1 formed a cellular border pattern, while in BMP4-treated cells, ZO-1 distribution appeared markedly affected, demonstrating that BMP4 disrupts ZO-1 organization in HRECs``.
- Figure 3 legend has been updated to ``Effect of BMP4 on ZO-1. Representative photographs of ZO-1 immunofluorescence (green) and nuclear stain (DAPI-Blue) in HRECs cells treated with BMP4 (50ng/ml) versus vehicle-treated controls. ZO-1 distribution appears to be disrupted in BMP4-treated HRECs in comparison to the normal distribution pattern in vehicle-treated HRECs``.
- In pg. 10, ln 309-310, the authors mentioned that MMPs are key players in REC “dysfunction in diabetic retinopathy.” Please elaborate on the roles of MMPs beyond dysfunction.
Response
- Thanks for your suggestion, in revision, more details have been MMP. Lines 397 and 406 read; The extracellular matrix (ECM) is known to play a supporting role for different tissues and also contributes to different biological functions, including regulation of the cell cycle and cell motility, and apoptosis. The ECM is made up of several molecules, including collagen, elastin and adhesion proteins, such as fibronectin and laminin; as well as proteases called matrix metalloproteases (MMPs) (Ref# 54). The MMPs belong to endopeptidases family that contains contain zinc and can degrade and remodel the ECM`s proteins (Ref# 55). MMPs family consists of 23 members which are divided based on their sub-cellular distribution and specificity for components of the ECM. The MMPs are divided into membrane-type matrix metalloproteases (MT-MMPs), collagenases, stromelysins, matrilysins and gelatinases members (MMP-2 and MMP-9) (Ref# 34).
- The Discussion section could be more in depth.
Response
In the revision, more details have been added to the discussion section (Lines 348-350; lines 385-390 and 394-406). Also, new references have been added (Ref#34, 36, 37,38, 44, 46-48, 51-55).

Reviewer 2 Report
The manuscript titled “Bone Morphogenetic Protein-4 Impairs Retinal Endothelial 1 Cell Barrier, A Potential Role in Diabetic Retinopathy” investigated the contribution of BMP-4 in the pathogenesis of diabetic retinopathy. The authors found increased levels of BMP-4 in the retinas of diabetic humans and 6-month diabetic mice. Using in vitro studies, the authors have shown that BMP-4 is involved in endothelial cell dysfunction and investigated the underlying molecular mechanisms. Overall, the manuscript is well-written overall, and the findings are interesting. However, there are some concerns that need to be addressed.
Comments:
1. The authors should provide information on the sample size of human and mouse retinas used in the materials and methods section. Additionally, regarding the postmortem human retina samples, it would be helpful to include details such as age and whether diabetic individuals had diabetic retinopathy.
2. Please clarify how the authors determined the concentration of BMP-4 (50 ng/mL) used in this study and the 48-hour duration in some of the experiments. If the choice was based on other studies, please provide the appropriate references.
3. The main concern revolves around the lack of correlation between the results obtained from retinal samples and in vitro experiments. The authors should provide data showing increased permeability of the inner blood-retinal barrier in mice using methods such as BSA immunostaining or Evans blue permeability assays. Moreover, the quality of IB4 staining in post-mortem human retinas in Fig. 1 appears poor, as it seems to be present all over the retina, and BMP-4 does not seem to be only in the retinal vessels. Additionally, the authors should present immunostaining for BMP-4 and IB4 separately, in addition to merged images. Furthermore, in Fig. 1B, a scale bar is missing, and statistical differences should be indicated in the figure legends.
4. In Fig.2B, the signal of IB4 staining appears scarce and should be improved. It is also necessary to add a scale bar in one of the images, and include statistical differences in the figure legends.
5. The graphs illustrating the fold change in protein levels should be normalized to the control, with the control set as 1. Therefore, the Y-axis in Fig 2A should indicate “Mean fold-change in BMP-4 normalized to b-actin” instead of “BMP/bActin”.
6. It would be valuable for the authors to include data on cell viability to ensure that the observed changes in permeability are not solely due to a decrease in cell viability resulting from incubation with BMP-4. Cell viability assays should be performed and the results presented in the manuscript.
7. In the Materials and Methods section, please indicate the specific vehicle that was used. Additionally, in the legend of Figure 5, it would be clearer to write "**P<0.01; ***P<0.001 versus vehicle-treated control HRECs" instead of using " ** = P < 0.01; *** = P < 0.001".
8. The rationale for using noggin (an antagonist of BMP-4), inhibitors of VEGFR2, p38, endoplasmic reticulum, or ERK should be better explained in the manuscript. Please provide more detailed information on why these inhibitors were chosen and their relevance to the study.
9. It would be helpful to provide improved images of control conditions showing ZO-1 immunostaining in a confluent monolayer, to better compare with experimental conditions.
10. Cell viability assays should be performed to confirm that the combination of rhBMP-4 and inhibitors of SU5416 (10 μM), PBA (30 μmol/l), SB202190, or U0126 (10μM) do not induce cell death. Moreover, the authors should provide images with the immunostaining of tight junction proteins to complement the permeability data.
11. The rationale for assessing MMPs activity in this study should be improved and better explained in the manuscript, as it does not seem to be directly related to the evaluated parameters and the discussion of the results.
12. In the first paragraph of the Discussion section, the authors claim to provide the first evidence for increased levels of BMP4 in the retinas of diabetic humans and mice compared to their control groups. However, a recent paper by Dong et al. (2021) shows upregulation of BMP-4 in retinas from STZ-induced diabetic rats (PMID: 33188599), which should be mentioned and discussed in relation to the current study.
13. It is recommended to include a western blot for a nuclear marker, such as Laminin A, to confirm the purity of nuclear extracts.
Author Response
Comments and Suggestions for Authors
Allow me to be microsurgically precise in this case, as a clinician and practitioner, and as a reviewer:
- the introduction is written impeccably and without repetition of sentence formulations known to me
- material and methods, both in humans and mice, described in detail and without omissions, as well as BMP4 assessment methods and in vitro study methods as well as MMP measurement methods
- results presented in textual, graphical and tabular form very convincingly and transparently with impressive photos of immunofluorescent staining techniques
- I have no objections to the statistical processing of the results
- the conclusion precisely states that the role of BMP4 in retinal endothelial cell dysfunction could be via activating both the canonical and noncanonical pathways and that BMP4 as a novel player that contributes to endothelial dysfunction in DR and shows us new targets for future therapy!
The reviewer's conclusion: I have no objections to the possible publication of this research, in fact, I recommend the publication!
Response
Thanks so much for your response

Reviewer 3 Report
Allow me to be microsurgically precise in this case, as a clinician and practitioner, and as a reviewer:
- the introduction is written impeccably and without repetition of sentence formulations known to me
- material and methods, both in humans and mice, described in detail and without omissions, as well as BMP4 assessment methods and in vitro study methods as well as MMP measurement methods
- results presented in textual, graphical and tabular form very convincingly and transparently with impressive photos of immunofluorescent staining techniques
- I have no objections to the statistical processing of the results
- the conclusion precisely states that the role of BMP4 in retinal endothelial cell dysfunction could be via activating both the canonical and noncanonical pathways and that BMP4 as a novel player that contributes to endothelial dysfunction in DR and shows us new targets for future therapy!
The reviewer's conclusion: I have no objections to the possible publication of this research, in fact, I recommend the publication!
Author Response
Reviewer 3
The manuscript titled “Bone Morphogenetic Protein-4 Impairs Retinal Endothelial 1 Cell Barrier, A Potential Role in Diabetic Retinopathy” investigated the contribution of BMP-4 in the pathogenesis of diabetic retinopathy. The authors found increased levels of BMP-4 in the retinas of diabetic humans and 6-month diabetic mice. Using in vitro studies, the authors have shown that BMP-4 is involved in endothelial cell dysfunction and investigated the underlying molecular mechanisms. Overall, the manuscript is well-written overall, and the findings are interesting. However, there are some concerns that need to be addressed.
Comments:
- The authors should provide information on the sample size of human and mouse retinas used in the materials and methods section. Additionally, regarding the postmortem human retina samples, it would be helpful to include details such as age and whether diabetic individuals had diabetic retinopathy.
Response:
Thank you so much for your suggestion. In the revision, the materials and methods section has been updated. Lines 84-85 read `` We obtained postmortem human eyes of non-diabetic and diabetic retinopathy subjects from Georgia Eye Bank (Atlanta, GA, USA) and Eversight (Ann Arbor, MI, USA) (40-60 years old, n=4-6 per group). ``. Lines 93-94 read now `` Wild-type (C57BL/6) and Akita mice were purchased from Jackson Laboratories (Bar Harbor, ME) (n=6-7 per group) ``.
- Please clarify how the authors determined the concentration of BMP-4 (50 ng/mL) used in this study and the 48-hour duration in some of the experiments. If the choice was based on other studies, please provide the appropriate references.
Response
- Yes, the choice of BMP4 (50 ng/ml) for 48 hours was based on our previous experience. Our previous data has shown that BMP4 reduced TER of ARPE-19 cells in a dose dependent manner compared to the vehicle-treated control cells over the experimental period. We reported a significant drop in TER of ARPE-19 by BMP4-treatmet in a dose dependent manner that started at 50 ng/mL.
- In the revision; we stated it clearly with our references. Lines 170-172 read `` The choice of BMP4 (50 ng/ml) for 48 hours was based on our previous experience in which we had a consistent and optimal effect at this concentration (Ref#12, 23 and 24).
- The main concern revolves around the lack of correlation between the results obtained from retinal samples and in vitro experiments. The authors should provide data showing increased permeability of the inner blood-retinal barrier in mice using methods such as BSA immunostaining or Evans blue permeability assays. Moreover, the quality of IB4 staining in post-mortem human retinas in Fig. 1 appears poor, as it seems to be present all over the retina, and BMP-4 does not seem to be only in the retinal vessels. Additionally, the authors should present immunostaining for BMP-4 and IB4 separately, in addition to merged images. Furthermore, in Fig. 1B, a scale bar is missing, and statistical differences should be indicated in the figure legends.
Response:
We and others have already established the permeability changes in diabetic mice in comparison to control (``A novel interaction between soluble epoxide hydrolase and the AT1 receptor in retinal microvascular damage. Wang & Al-Shabrawey, M. (2020). A Prostaglandins & other lipid mediator, 148, 106449``, ``A lipidomic screen of hyperglycemia-treated HRECs links 12/15-Lipoxygenase to microvascular dysfunction during diabetic retinopathy via NADPH oxidase. Ibrahim, A. S.... & Al-Shabrawey, M. (2015). Journal of lipid research, 56(3), 599-611``. Also, Ran et. al reported that STZ diabetic animal model has shown significantly increased in the BRB permeability using albumin-bound EB when compared with that in the normal controls (P<0.01)).
The goal of this study was mainly to characterize the changes in retinal levels of BMP4 in diabetic mice and testing the direct effect on retinal endothelial cell barrier. Our current findings set the ground for future studies to correlate the changes in BMP4 levels with different stages of diabetic retinopathy.
- In the revision, (discussion section) Lines 394-397 read ``Regarding the retinal-vascular microenvironment, We and others have already established the permeability changes in diabetic mice in comparison to control using BSA immunostaining or Evans blue permeability assays compared with that in the normal controls (Ref# 31, 51-53)``.
- Regarding Fig1. We agree with the reviewer that IB4 staining in post-mortem human retinas appears poor. We noticed that IB4 was not perfect with human paraffin-embedded sections. We included a higher resolution figure with scale bar in the revision.
- In Fig.2B, the signal of IB4 staining appears scarce and should be improved. It is also necessary to add a scale bar in one of the images, and include statistical differences in the figure legends.
Response
- Thank you so much for your suggestion. In the revision Figure 2 has been updated.
- The graphs illustrating the fold change in protein levels should be normalized to the control, with the control set as 1. Therefore, the Y-axis in Fig 2A should indicate “Mean fold-change in BMP-4 normalized to b-actin” instead of “BMP/bActin”.
Response
- Thank you so much for your suggestion. In the revision Figure 2 has been updated.
- It would be valuable for the authors to include data on cell viability to ensure that the observed changes in permeability are not solely due to a decrease in cell viability resulting from incubation with BMP-4. Cell viability assays should be performed and the results presented in the manuscript.
Response
- Thank you so much for this good observation. In this study, we focused on studying the barrier function. Here, HRECs were grown in 96-well electrode arrays. This array uses a gold electrode substrate beneath the cells. As the cells grow, the resistance increase. The confluent mature monolayers of HRECs have been confirmed by the plateau phase of electric resistance. To make sure that the effect of drugs (as BMP4 here) is due to barrier function not due the cell death, we present the data as normalized TER resistance calculated by dividing the resistance of each well measured in ohms at each time point by the baseline resistance (ohms) obtained before the adding of the BMP4 and plotted as a function of time. On the other hand, ECIS provides real time information of cell capacitance that reflects the cell survival and adherence throughout the entire period of the experiment. We explained this further in the method section (Lines 180-182).
- However, we are planning for a large-scale study to assess the different effects of BMP4 on cell machinery.
- In the Materials and Methods section, please indicate the specific vehicle that was used. Additionally, in the legend of Figure 5, it would be clearer to write "**P<0.01; ***P<0.001 versus vehicle-treated control HRECs" instead of using " ** = P < 0.01; *** = P < 0.001".
Response
- In revision, we updated the materials and methods. Lines 168-170 read `` The BMP was prepared by dissolving in 4 mM HCl (according to the manufacture) and then diluted with the media to get out final concentration``
- The legend of Figure 5 has been updated.
- The rationale for using noggin (an antagonist of BMP-4), inhibitors of VEGFR2, p38, endoplasmic reticulum, or ERK should be better explained in the manuscript. Please provide more detailed information on why these inhibitors were chosen and their relevance to the study.
Response
In the revision, more details have been added
- In the Material and Methods section (Assessment of human retinal endothelial cell barrier function), lines 157-161 read ``The Vascular endothelial growth factor (VEGF) and bone morphogenetic proteins (BMPs), are key regulators for the BRB. several studies have shown that BMP4 mediate its effects through VEGF-A/VEGFR2 and angiopoietin-1/TIE2 signaling stimulation (Ref#25 and 26). On the other hand, BMPs signaling was regulated by different regulating factors such as noggin, chordin, and gremlin (Ref#27)``.
- In the discussion section, lines 385-390 read ``In addition, many studies have reported the cross-talk between BMP4 and VEGF. VEGF has been shown to induce BMPs expression in the microvascular endothelial cells, including BMP2 and BMP4 (Ref# 45 and 46). VEGF has been recognized as a downstream target from smad and p38 MAPK (Ref# 35 and 47). VEGF and BMP4 have been shown to enhance the high expression of PECAM1 and VE-cadherin, which play an important role in endothelial cell permeability and the development of new blood vessels (Ref# 37, 45 and 48).
- It would be helpful to provide improved images of control conditions showing ZO-1 immunostaining
in a confluent monolayer, to better compare with experimental conditions.
Response
- In the revision, improved images of control condition have been added.
- Cell viability assays should be performed to confirm that the combination of rhBMP-4 and inhibitors of SU5416 (10 μM), PBA (30 μmol/l), SB202190, or U0126 (10μM) do not induce cell death. Moreover, the authors should provide images with the immunostaining of tight junction proteins to complement the permeability data.
Response
- Our preliminary data discuss the effect of BMP4 alone and in combination of different inhibitors on the barrier function. However, we are planning for a large-scale study to assess the different effects of BMP4 on cell machinery.
- Regarding to the cell viability, please refer to our response to comment #6.
- The rationale for assessing MMPs activity in this study should be improved and better explained
in the manuscript, as it does not seem to be directly related to the evaluated parameters and the
discussion of the results.
Response
- Thanks for your suggestion, in revision, more details have been MMP. Lines 397 and 406 read; The extracellular matrix (ECM) is known to play a supporting role for different tissues and also contributes to different biological functions, including regulation of the cell cycle and cell motility, and apoptosis. The ECM is made up of several molecules, including collagen, elastin and adhesion proteins, such as fibronectin and laminin; as well as proteases called matrix metalloproteases (MMPs) (Ref# 54). The MMPs belong to endopeptidases family that contains contain zinc and can degrade and remodel the ECM`s proteins (Ref# 55). MMPs family consists of 23 members which are divided based on their sub-cellular distribution and specificity for components of the ECM. The MMPs are divided into membrane-type matrix metalloproteases (MT-MMPs), collagenases, stromelysins, matrilysins and gelatinases members (MMP-2 and MMP-9) (Ref# 34).
- In the first paragraph of the Discussion section, the authors claim to provide the first evidence for
increased levels of BMP4 in the retinas of diabetic humans and mice compared to their control groups.
However, a recent paper by Dong et al. (2021) shows upregulation of BMP-4 in retinas from STZ-2
induced diabetic rats (PMID: 33188599), which should be mentioned and discussed in relation to the
current study.
Response
- Thank you so much. The 1st paragraph of the discussion section has been updated. Lines 348-350 read ``In consistent with our results, Dong group has reported that BMP4 and SMAD9 were highly expressed in STZ-induced diabetic rats (~2-fold) in comparison to control group (Ref# 38)``.
- It is recommended to include a western blot for a nuclear marker, such as Laminin A, to confirm the purity of nuclear extracts.
Response
- Thank you so much for you suggestion. We agree with you. In this study we used HDAC1 which play important roles in transcriptional regulation in eukaryotic cells and commonly used as a nuclear extract marker for WB studies. We are planning to do more molecular study to understand the deep molecular response and evaluate the multiple function of BMP4 and will consider your suggestion in our future experiments.

Round 2
Reviewer 1 Report
There is no further comment from this reviewer.
Reviewer 2 Report
The authors have improved the manuscript. It is well-organized, clearly written, and suitably discussed.